# Cryoablation and Immunotherapy: An Enthralling Synergy for Cancer Treatment

Zain al Abidine Medlej [1,*], Wassim Medlej [2], Sami Slaba [3], Pedro Torrecillas [4], Antonio Cueto [4], Alberto Urbaneja [4], Adolfo Jimenes Garrido [4] and Franco Lugnani [4]

[1] Agro-Food and Environmental Biosciences and Technologies Department, University of Teramo, 64100 Teramo, Italy
[2] Cryolebabon and Medical Devices Sarl, Beirut 1107 2020, Lebanon; wassim@cryoleb.com
[3] Hotel Dieu de France Hospital, Saint-Joseph University, Beirut 1104 2020, Lebanon; sami.slaba@hdf.usj.edu.lb
[4] Radiology Department, Clinica Santa Elena, 29620 Madrid, Spain; doctortorrecillas@criocirugiainternacional.com (P.T.); acueto@criocirugiainternacional.com (A.C.); aurbaneja@criocirugiainternacional.com (A.U.); ajimenez@criocirugiainternacional.com (A.J.G.); franco@lugnani.com (F.L.)
[*] Correspondence: zaamedlej@unite.it

**Abstract:** As less invasive options for surgical tumor removal, minimally invasive ablative techniques have gained popularity. Several solid tumors are being treated with cryoablation, a non-heat-based ablation technique. Cryoablation data in comparison over time demonstrates better tumor response and faster recovery. Combining cryosurgery with other cancer therapies has been explored to improve the cancer-killing process. Cryoablation with the combination of immunotherapy, results in a robust and efficient attack on the cancer cells. This article focuses on investigating the ability of cryosurgery to create a strong antitumor response when combined with immunologic agents resulting in a synergetic effect. To achieve this objective, we combined cryosurgery with immunotherapy using Nivolumab and lpilimumab. Five clinical cases of lymph node, lung cancer, bone, and lung metastasis were followed and analyzed. In this series of patients, percutaneous cryoablation and addressing immunity agents were technically feasible. In the follow-ups, there appeared to be no radiological evidence of new tumor development.

**Keywords:** immunity; cryosurgery; cancer

## 1. Introduction

Cancer is the second leading cause of death globally [1,2]. Immunotherapy, which uses the immune system to fight cancer, has emerged as an effective and possibly revolutionary method of treating cancer and cancer metastasis [3,4].

The immunosuppressive tumor microenvironment (TME), for instance immunologically cold conditions, including malignant cells and deactivated or weakened immune cells, is a major contributor to the subpar clinical results of immunotherapy of malignancies [5].

Cryosurgery involves freezing a tumor to the point where ice forms (frostbite), which results in cryoinjury and cancer cell death within the frozen tissue ice ball [6–9]. As a result, tumor antigens may be released, and damage-associated molecular patterns (DAMPs) may be created [10–13]. These substances have the potential to trigger immunogenic cell death (ICD), create an immuno-active (i.e., immunologically hot) TME, and then activate dendritic cells (DCs), macrophages, and CD8+ cytotoxic T lymphocytes (CTLs) to carry out anticancer immunotherapy [11,13–16].

Additionally, in contrast to high-temperature thermal therapy, which causes vascular stasis during heating, there is a brief (a few hours) reperfusion of the tumor after thawing a frozen tumor ice ball [17–19], which may facilitate effective immune cell infiltration into the tumor after cryosurgery.

However, cryosurgery by itself is insufficient to trigger an effective immunotherapeutic impact against cancer. However, while the peripheral region of a frozen tumor ice ball observable by medical ultrasonography has a temperature over −20 °C and up to 4 °C, a temperature of −20 °C or lower is required to guarantee cell death [9,20–22]. The latter is because during cryosurgery, biological tissues gradually freeze as temperature decreases starting at −0.6 °C and continuing until below −20 °C [22,23] and the extent of ice formation is insufficient for ultrasonography to detect until −4 °C. Thus, neither ice formation above −4 °C nor the location of the temperature below −20 °C required to assure cancer cell death can be determined by ultrasonography. This might result in inadequate tumor removal and post-cryosurgery cancer recurrence [6,12,24,25].

As a result, the possibility of combining cryosurgery with other cancer treatments has been investigated to enhance the therapeutic outcome by increasing cancer-killing in the peripheral region of a frozen tumor ice ball [26–33].

Immunotherapy boosts a person's immune system to recognize and combat cancer cells. Cryoablation is thought to have increased expression of tumor-specific antigens, which the body can recognize as foreign invaders and, when combined with immunotherapy, make the body's attack on cancer cells even more potent and effective.

This study investigated the ability of cryoablation to create a robust antitumor response in combination with immunologic agents. The current study hypothesized that this combination would result in a synergistic effect against cancer cells. To the best of our knowledge, this is the first study to examine the combination of cryotherapy and immunologic agents in different cancer types as lymphatic nodules, small cell lung cancer in females and males, lung cancer with bone metastases, bilateral lung metastasis, and mediastinal lymph nodes.

## 2. Overview of the Current Ablation Techniques

There are various ablation modalities but the most common are Radiofrequency Ablation (RFA), Microwave ablation (MWA), and percutaneous cryoablation (PTA), also called cryosurgery, cryotherapy, or cryoablation.

### 2.1. Radiofrequency Ablation (RFA)

RFA, which was first used to treat hepatic lesions, was formerly the most used ablation procedure for treating renal cell carcinoma (RCC). This technique is done through a radiofrequency electrode which can have different types and shapes using high-frequency alternating current (460–500 kHz). This causes ionic agitation, which leads to frictional heat (temperatures of 60–100 °C) leading to cellular protein denaturation and coagulative necrosis [34]. Temperatures at the electrode–tissue interface are decreased by enabling cooling fluid to circulate inside the electrode, reducing charring [35].

RFA is conveniently accessible, inexpensive, and has a shorter ablation duration than the PTA [36]. If the lesion has a diameter greater than 3 cm, RFA has a lower success rate. Another downside of RFA is heat dissipation from the top of the electrode, which can result in damage to nearby structures [37].

### 2.2. Microwave Ablation (MWA)

MWA produces coagulative necrosis by emitting electromagnetic waves with a frequency exceeding 900 MHz [38]. An MWA probe disrupts water molecules, causing heat dissipation, cellular death, and tissue damage. MV ablation reaches a higher temperature than other heat-based procedures, which may be favorable since it causes more intralesional damage and may aid in the treatment of larger lesions (diameter > 3 cm) in less time [39]. Moreover, MWA lacks the heat sink effect. However, MWA is more painful compared with the other ablation techniques. As well, MWA results are more dangerous in hilar lesions.

### 2.3. Percutaneous Cryoablation (PTA)/Cryotherapy/Cryoablation/Cryosurgery

Cryoprobes are typically connected to Argon, which may rapidly achieve temperatures of around −180 degrees Celsius. This is accomplished by rapidly expanding a high-pressure gas through a valve. While the gas is insulated, a rapid cooling effect is produced, resulting in no heat transmission to the surrounding environment. An ice ball forms at the end of a cryoprobe as a result of the rapid cooling. The freezing and thawing process disrupts the cell membrane and causes microvascular injury. This generates hypotonic stress, which induces necrosis in cells. Multiple probes can be utilized simultaneously at a minimum distance of 1–2 cm [40]. Cryoablation allows for the treatment of larger lesions, and if the diameter is greater than 3 cm, the oncological outcomes are superior to RFA [41]. Furthermore, with larger lesions, the damage rate is lower near the collecting system [42]. Another advantage of cryoablation over heat-based ablation is that it is less painful, as the cold acts as an anesthetic. Cryoablation, on the other hand, takes longer than RFA due to the freezing-thawing cycles, and the ice ball does not have a uniform effect. Indeed, the extremely low temperature achieved inside the ice ball causes damage, although the zero degrees achieved on the lesion's periphery do not result in cellular tissue death [43]. Because surrounding blood vessels are not directly cauterized as in RFA, cryoablation is associated with a higher risk of bleeding.

## 3. Materials and Methods

The paper is organized as follows. In Section 3, we briefly report the methodological approach, and we present the different cases with the relative therapy. Section 4 presents the results. In Section 5, we discuss the results. The conclusions are presented in Section 6. Finally, in Section 7 we examine some research limitations.

### 3.1. The Cryoablation Procedure

Remarkably extended in the 1990s, cryosurgery is being used for treating benign and malignant tumors. It was shown that cryosurgery is a great technique for treating non-resectable cancers, especially those which are metastatic [44]. Clinically, Cryosurgical methods are based on widely acknowledged scientific theories that support physician-managed destruction of localized tumors [6,8]. Cryosurgery, which is carried out using multiprobe devices and advanced imaging techniques, has produced successful short-term outcomes in the treatment of cancer [8].

Generally, once the diagnosis of one or more tumors in the patient has been made and the line of intervention has been decided, the cryosurgical treatment is carried out under partial or total anesthesia. Diagnosis is made using CT images, PET images, MRIs, and/or ultrasounds to ensure that the tumor can be identified.

On the day of the intervention, the site is marked on the skin. In the case of local anesthesia, the latter is injected into the targeted skin and the area around the tumor. The cancer is measured to estimate how long it takes to freeze.

Once the area is numb, a small incision will be made to insert the cryoprobe, which is placed through the tumor with ultrasound and/or CT guidance. If the cancer is large (more than 2 cm), multiple cryoprobes will be inserted.

In the cryoprobes, measuring 1.7 mm, 2.4 mm, or 3.8 mm in diameter (depending on the case) argon gas will be used and delivered from a separate computerized device from cylinders, are inserted into the center and the periphery of the tumor mass, if the latter is larger than 5 cm in diameter. The peripheral insertion of the cryoprobes should not be distant more than 1 cm from the margin of the tumor mass.

The preoperative local anesthesia is carried out together with or without sedation accordingly to patient compliance and the difficulty of the procedure.

In general, the freezing time is 10 min, while the thawing time is 5 to 15 min.

In the freezing phase, argon will pass through the probe to allow the adjacent tissue to freeze. The argon gas in the probes reaches a temperature of approximately −180 °C at the tip of the probe and causes cell death within the −40 °C isotherm located approximately

1 cm inside the margin of the ice ball, while the periphery of the ice ball grows close to water freezing temperature. This is the reason why the tumor itself and an additional margin of tissue around it are treated. The exact cycle time is adjusted on the size of the lesion being treated. The cooling cycle is repeated a second time, with a similar procedure, to increase the effectiveness of the first cycle. Then the lesion is thawed passively. At the end of the procedure, the probe is heated with a flow of Helium gas to facilitate the removal of the cryoprobe stuck by ice inside the lesion. The cancer cells die at the end of the procedure and in the weeks following and are reabsorbed resulting in scar tissue.

*3.2. Clinical Cases Description*

We describe the tumor types relative to the clinical cases that we will be presented in the successive section. The cancer types are the following: lymphatic nodules, small cell lung cancer (SCC), and bilateral lung and mediastinal lymph node metastases.

### 3.2.1. Lymphatic Nodules

The swelling of the lymph nodes in the thorax, particularly the mediastinum, is known as mediastinal lymphadenopathy. The emergence of mediastinal lymphadenopathy indicates the presence of an illness or infection [45]. There are a variety of reasons for mediastinal lymphadenopathy, some of which affect the lung and others which affect the entire body. Cancer metastasis, coccidioidomycosis, chronic obstructive pulmonary disease, cystic fibrosis, and esophageal and lung cancers are some of the causes, according to [45]. Mediastinal lymphadenopathy can frequently be detected by diagnostic imaging of lymph nodes using computed tomography (CT) or positron emission tomography (PET) [46]. A biopsy can be conducted to inspect the cells if cancer is thought to be present.

### 3.2.2. Lung Cancer

The incidence of lung cancer is continuously increasing and remains the primary cause of death from cancer in both men and women [47]. Lung cancer can be divided into two categories: small-cell lung cancer (SCC) as well as non-small cell lung cancer (NSCC). The non-small cell lung cancer (NSCC) is divided into three types: squamous cell carcinoma, adenocarcinoma, and large cell carcinoma. Possible treatments for NSCC are surgery, radiation therapy, chemotherapy, thermal ablation, or a combination of these therapies. On the other hand, small cell lung cancer (SCC) is common in clinical practice. SCC is usually inoperable because it is often widespread at the time of diagnosis. Cryoablation is a technique used to treat these neoplasms [48].

### 3.2.3. Bilateral Lung Metastasis and Mediastinal Lymph Nodes (Near the Esophagus)

Lung metastases represent distant localizations of a malignant tumor in the lung (one of the most frequent sites for the spread of tumor diseases). Potentially all tumors can metastasize to the lungs, some more frequently, some less. Metastasis occurs via blood, i.e., through the bloodstream. Tumors drain blood into the vena cava where they find their first potential site of spread in the lungs [49]. Normally patients with lung metastases, who cannot resort to surgery, are more often with a poor prognosis. These are cases in which chemotherapy or radiotherapy treatments often become palliative or which can make the disease chronic, but cannot find cure margins [50]. The study by [51] pointed out that lung metastasis can be cured by cryoablation. Diagnostic imaging of metastases with computed tomography (CT) or positron emission tomography (PET) can often identify the metastasis.

*3.3. Clinical Cases Presentation*

We present five clinical cases that were treated in our unit "Cryosurgery Center" department at the "Hotel Dieu de France" hospital in Beirut, Lebanon, and Clinica Santa Elena, Spain. The first clinical case is related to lymphatic nodules, and the second and third regard small cell lung cancer (SCC) in different genders, respectively. The fourth clinical

case concerns lung cancer with bone metastases while the fifth corresponds to bilateral lung and mediastinal lymph node metastases.

### 3.3.1. Lymphatic Nodules

In March 2020, a 35-year-old patient arrived at our department with the results of a recently performed positive biopsy. For this reason, a CT scan of the chest, abdomen, and pelvis with means of contrast was performed in our department. CT examination confirmed the presence of several mediastinal lymph nodes. An anterior mediastinal lymph node with a diameter of (39 × 26 mm), a para-aortic lymph node with a diameter of (35 × 25 mm), a lymph node of the aorta-pulmonary window with a diameter of (30 × 20 mm) as well as several lymph nodes with diameters ranging from (27 × 16 mm) to (23 × 20 mm). Focal pleural thickening of 10 mm was confirmed at the posterior right lower lobe (peri-vertebral) (see Figure 1). For these reasons, it was decided to subject the patient to a cryosurgery intervention procedure as described above.

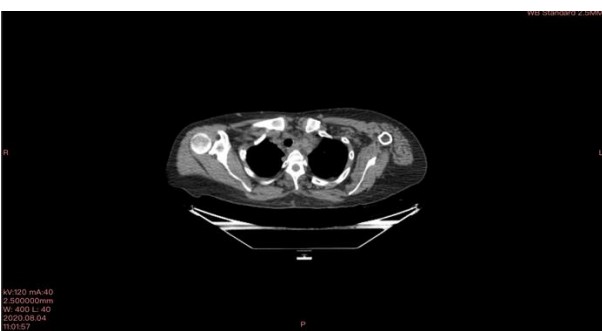

**Figure 1.** CT scan image executed.

Directly after the cryosurgery intervention, specifically, after the ice melting, Nivolumab (Opdivo, New York City, NY, USA) (100 mg/10 mL) is directly given as 2 mg as an intravenous infusion into the tumor, accompanied by Ipilimumab (Yervoy, New York City/ U.S.) 50 mg/10 mL) given as well as 2 mg as an intravenous infusion into the tumor.

Furthermore, the patient underwent an immunotherapy treatment every 21 days for a duration of 6 months using Nivolumab (Opdivo) (100 mg/10 mL), Ipilimumab (Yervoy) (50 mg/10 mL), and Nivolumab (Opdivo) 240 mg. The Nivolumab (Opdivo) (100 mg/10 mL) is administered with a dosage of 1 mg/kg as an intravenous infusion over 30 min. Subsequently, the patient's body should rest for 30 min. Afterward, Ipilimumab (Yervoy) (50 mg/10 mL) is administered as 3 mg/kg through intravenous infusion for 1 h and 30 min. The treatment should be repeated every 3 weeks for up to 6 doses or until unacceptable toxicity. Moreover, every 2 weeks Nivolumab (Opdivo) 240 mg should be administered via intravenous infusion for 6 months.

### 3.3.2. Lung Cancer

We present two cases of end-stage small cell lung cancer (SCC) treated with cryosurgery and local immunotherapy.

#### Small Cell Lung Cancer (SCC)—Female Patient

In April 2020, a 59-year-old smoker patient visited our department due to dyspnea and dysphagia.

The patient's history and physical examination as well as recent imaging evaluation should be well evaluated. Chest CT is the key imaging modality for evaluating lung tumor size and the location of lesions. Because of the complaint also of dysphagia, we conducted a CT scan of the chest, abdomen, and pelvis with means of contrast. A biopsy was taken as advised by [52]. The presence of small cell lung cancer, stage IVA (T4 N3 M1a) was confirmed. PET-CT and CT scans show images of a hypermetabolic left pulmonary hilar tumor (Figure 2).

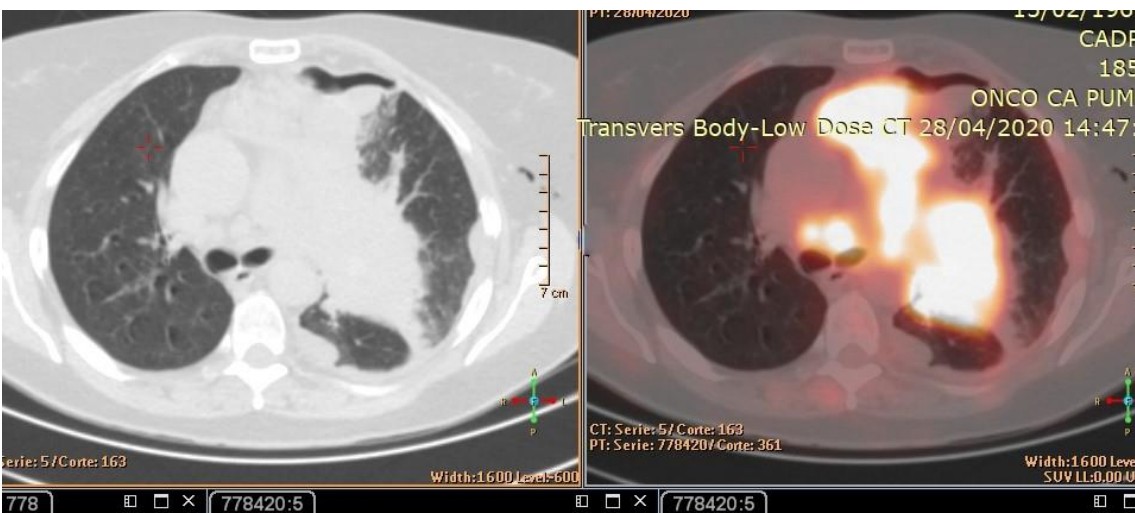

**Figure 2.** CT scan (**left**) and Pet-CT (**right**) images of hypermetabolic left pulmonary hilar tumor from a female SCC patient.

After performing the necessary blood tests, the date of the operation was confirmed under local anesthesia and sedation. Drainage of 1500 cc of pleural fluid was accomplished and cryoablation of the iliomediastinal tumor was performed with two freeze-thaw cycles using eight probes of cryoablation. Finally, cryosurgery is completed with an intratumor injection of the monoclonal antibody Ipilimumab (Yervoy) and Nivolumab (Opdivo).

Directly after the cryosurgery intervention, specifically, after the ice melting, Nivolumab (Opdivo) (100 mg/10 mL) is directly given as 2 mg as an intravenous infusion into the tumor, accompanied by Ipilimumab (Yervoy) (50 mg/10 mL) given as well as 2 mg as an intravenous infusion into the tumor.

Furthermore, the patient underwent an immunotherapy treatment every 21 days for a duration of 6 months using Nivolumab (Opdivo) (100 mg/10 mL), Ipilimumab (Yervoy) (50 mg/10 mL), and Nivolumab (Opdivo) 240 mg. The Nivolumab (Opdivo) (100 mg/10 mL) is administered with a dosage of 1 mg/kg as an intravenous infusion over 30 min. Subsequently, the patient's body should rest for 30 min. Afterward, Ipilimumab (Yervoy) (50 mg/10 mL) is administered as 3 mg/kg through intravenous infusion for 1 h and 30 min. The treatment should be repeated every 3 weeks for up to 6 doses or until unacceptable toxicity. Moreover, every 2 weeks Nivolumab (Opdivo) 240 mg should be administered via intravenous infusion for 6 months.

Small Cell Lung Cancer (SCC)—Male Patient

The patient, a 60-year-old man, was admitted to our Cryosurgery Unit in May 2020.

A CT scan and biopsy showed the presence of small cell lung cancer, stage IVA (T4 N3 M1a). The CT scan showed images of a hypermetabolic left pulmonary hilar tumor (Figure 3).

The necessary blood tests have been carried out. During cryotherapy, 1500 cc of pleural fluid was drained and cryotherapy was adopted to treat the iliomediastinal tumor with two freeze-thaw cycles.

Directly after the completion of the cryosurgery and after ice melting, Nivolumab (Opdivo) (100 mg/10 mL) is directly given at 2 mg as an intravenous infusion into the tumor, accompanied by Ipilimumab (Yervoy) (50 mg/10 mL) given at 2 mg as an intravenous infusion into the tumor.

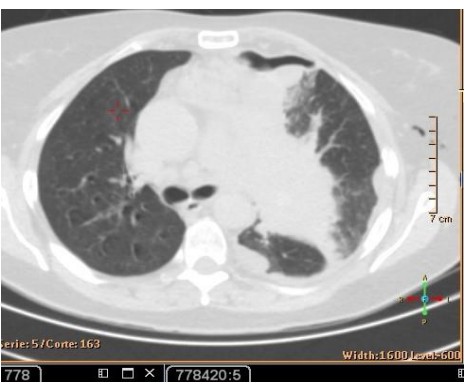

**Figure 3.** CT scan image showing hypermetabolic left pulmonary hilar tumor from male SCC patient.

Furthermore, the patient underwent an immunotherapy treatment every 21 days for a duration of 6 months using Nivolumab (Opdivo) (100 mg/10 mL), Ipilimumab (Yervoy) (50 mg/10 mL), and Nivolumab (Opdivo) 240 mg. The Nivolumab (Opdivo) (100 mg/10 mL) is administered with a dosage of 1 mg/kg as an intravenous infusion over 30 min. Subsequently, the patient's body should rest for 30 min. Afterward, Ipilimumab (Yervoy) (50 mg/10 mL) is administered as 3 mg/kg through intravenous infusion for 1 h and 30 min. The treatment should be repeated every 3 weeks for up to 6 doses or until unacceptable toxicity. Moreover, every 2 weeks, 240 mg of Nivolumab (Opdivo) should be administered via intravenous infusion for 6 months.

### 3.3.3. Lung Cancer with Bone Metastases

In July 2021, a 60-year-old patient came to our Cryosurgery unit, provided with a CT scan of the chest dated April 2021. The CT scan revealed the presence of a hyper-metabolic lesion with a size of 2.8 cm and involving the right upper/middle lobe related to the neoplastic process. The presence of hypermetabolic lytic bone lesions involving the right proximal femur T-10 and the left transverse process of T-9 was also confirmed, highlighting bone metastasis (Figure 4).

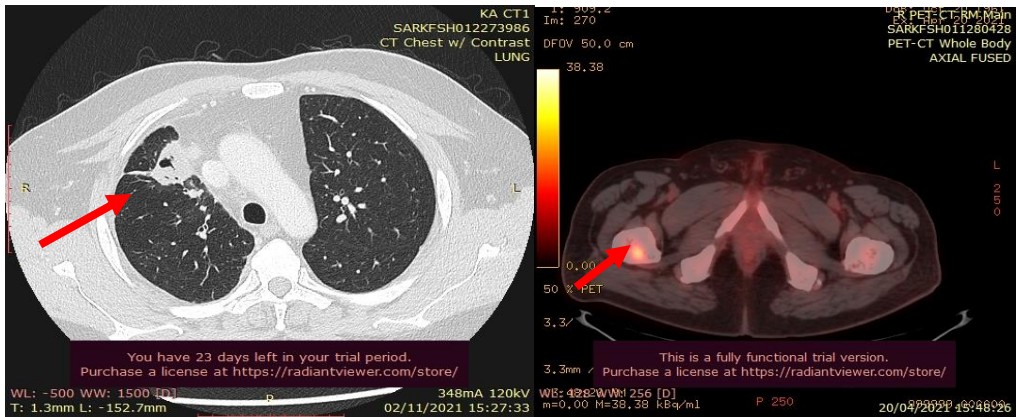

**Figure 4.** CT scan (**left**) and Pet- CT (**right**) images from April 2021 with evidence of bone metastases. The arrows indicate the tumor's location.

The process for patients with metastatic disease should include abdominal and pelvic CT scans. This is the reason why an abdominopelvic CT scan with contrast was performed in July 2021. There was no evidence of abdominopelvic metastases but always manifestations of diffusion of bone metastases. A specular nodule was noted in the right upper lobe with evidence of bilateral pulmonary nodules, and pleural and bone metastases. The pleural biopsy performed confirmed the presence of metastatic adenocarcinoma.

Subsequently, in August 2021, the patient underwent the operation of total mass destruction in the right upper lobe lung by cryosurgery where two 2.4 mm cryoprobes were used (Figure 5).

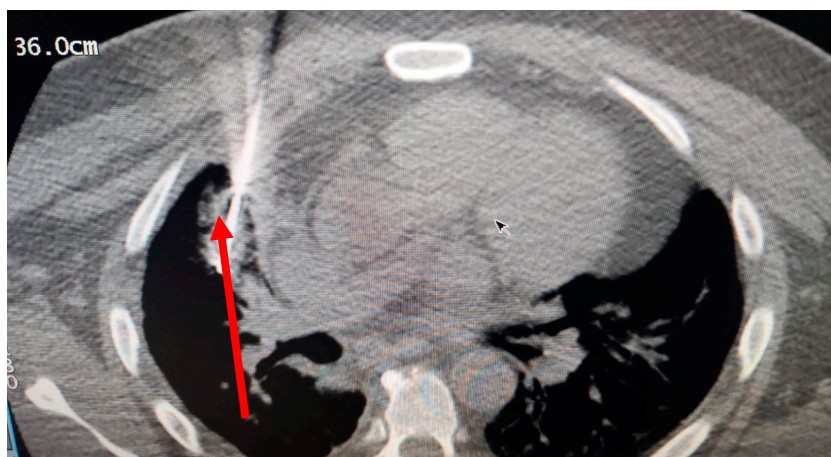

**Figure 5.** The probe inside the tumor, cryoablation date: 11 August 2021 at the cryosurgery center.

Directly after the cryosurgery intervention, specifically after the ice melting, Nivolumab (Opdivo) (100 mg/10 mL) is directly given at 2 mg as an intravenous infusion into the tumor, accompanied by Ipilimumab (Yervoy) (50 mg/10 mL) given at 2 mg as an intravenous infusion into the tumor.

Furthermore, the patient underwent an immunotherapy treatment every 21 days for a duration of 6 months using Nivolumab (Opdivo) (100 mg/10 mL), Ipilimumab (Yervoy) (50 mg/10 mL), and Nivolumab (Opdivo) 240 mg. The Nivolumab (Opdivo) (100 mg/10 mL) is administered with a dosage of 1 mg/kg as an intravenous infusion over 30 min. Subsequently, the patient's body should rest for 30 min. Afterward, Ipilimumab (Yervoy) (50 mg/10 mL) is administered at 3 mg/kg through intravenous infusion for 1 h and 30 min. The treatment should be repeated every 3 weeks for up to 6 doses or until unacceptable toxicity. Moreover, every 2 weeks 240 mg ofNivolumab (Opdivo) should be administered via intravenous infusion for 6 months.

3.3.4. Bilateral Lung Metastasis and Mediastinal Lymph Nodes (Near the Esophagus)

In October 2018, a 24-year-old patient arrived at our cryosurgery unit with the results of a CT scan. The presence of bilateral pulmonary and mediastinal lymph node metastases close to the esophagus was highlighted (Figure 6).

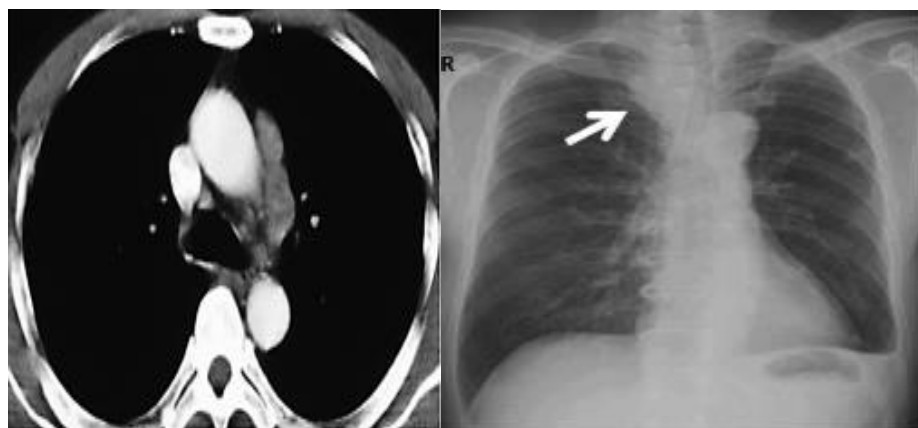

**Figure 6.** CT scan images indicate the presence of bilateral pulmonary and mediastinal lymph node metastases close to the esophagus. The arrow indicates the tumor's location.

On 25 October 2018, it was decided to perform a cryotherapy session at our department, under local anesthesia and sedation. Cryoablation was performed for 15 min using 3 probes in the right hemithorax on lung metastases: mediastinal and hilar localization.

Directly after the cryosurgery intervention, specifically after the ice melting, Nivolumab (Opdivo) (100 mg/10 mL) is directly given at 2 mg as an intravenous infusion into the tumor, accompanied by Ipilimumab (Yervoy) (50 mg/10 mL) given at 2 mg as an intravenous infusion into the tumor.

Furthermore, the patient underwent immunotherapy treatment every 21 days for a duration of 6 months using Nivolumab (Opdivo) (100 mg/10 mL), Ipilimumab (Yervoy) (50 mg/10 mL), and Nivolumab (Opdivo) 240 mg. The Nivolumab (Opdivo) (100 mg/10 mL) is administered with a dosage of 1 mg/kg as an intravenous infusion over 30 min. Subsequently, the patient's body should rest for 30 min. Afterward, Ipilimumab (Yervoy) (50 mg/10 mL) is administered at 3 mg/kg through intravenous infusion for 1 h and 30 min. The treatment should be repeated every 3 weeks for up to 6 doses or until unacceptable toxicity. Moreover, every 2 weeks, 240 mg of Nivolumab (Opdivo) should be administered via intravenous infusion for 6 months.

Table 1 below resume all the clinical cases.

**Table 1.** Schematization of the clinical studied cases.

| Cancer Type | Age | Gender | Admission Year | Results CT, PT, and/or Biopsy Prior Cryosurgery |
|---|---|---|---|---|
| Lymphatic nodules | 35 | Not indicated | 2020 | A- Presence of several mediastinal lymph nodes.<br>B- An anterior mediastinal lymph node with a diameter of (39 × 26 mm)<br>C- A para-aortic lymph node with a diameter of (35 × 25 mm)<br>D- A lymph node of the aorta-pulmonary window with a diameter of (30 × 20 mm)<br>E- Several lymph nodes with diameters ranging from (27 × 16 mm) to (23 × 20 mm).<br>F- Focal pleural thickening of 10 mm was confirmed at the posterior right lower lobe (peri-vertebral). |
| Small cell Lung cancer (SCC) | 59 | Female | 2020 | A- Hypermetabolic left pulmonary hilar tumor<br>B- The presence of small cell lung cancer, stage IVA (T4 N3 M1a) was confirmed after biopsy. |
| Small cell Lung cancer (SCC) | 60 | Male | 2020 | A- CT scan showed images of a hypermetabolic left pulmonary hilar tumor.<br>B- A CT scan and biopsy showed the presence of small cell lung cancer, stage IVA (T4 N3 M1a). |
| Lung cancer with bone metastases | 60 | Not indicated | 2021 | A- The CT scan revealed the presence of a hyper-metabolic lesion sizing 2.8 cm and involving the right upper/middle lobe related to the neoplastic process.<br>B- The presence of hypermetabolic lytic bone lesions involving the right proximal femur T-10 and the left transverse process of T-9 was also confirmed, highlighting bone metastasis.<br>C- There was no evidence of abdominopelvic metastases but always manifestations of diffusion of bone metastases.<br>D- A specular nodule was noted in the right upper lobe with evidence of bilateral pulmonary nodules, and pleural and bone metastases.<br>E- The pleural biopsy performed confirmed the presence of metastatic adenocarcinoma. |

**Table 1.** *Cont.*

| Cancer Type | Age | Gender | Admission Year | Results CT, PT, and/or Biopsy Prior Cryosurgery |
|---|---|---|---|---|
| Bilateral lung metastasis and mediastinal lymph nodes (near esophagus) | 24 | Not indicated | 2018 | A- The presence of bilateral pulmonary and mediastinal lymph node metastases close to the esophagus. |

| Immunotherapy (for all cases) |
|---|
| Directly after the cryosurgery intervention, specifically, after the ice melting, Nivolumab (Opdivo) (100 mg/10 mL) is directly given at 2 mg as an intravenous infusion into the tumor, accompanied by Ipilimumab (Yervoy) (50 mg/10 mL) given at 2 mg as an intravenous infusion into the tumor. Furthermore, the patient underwent an immunotherapy treatment every 21 days for a duration of 6 months using Nivolumab (Opdivo) (100 mg/10 mL), Ipilimumab (Yervoy) (50 mg/10 mL), and Nivolumab (Opdivo) 240 mg. The Nivolumab (Opdivo) (100 mg/10 mL) is administered with a dosage of 1 mg/kg as an intravenous infusion over 30 min. Subsequently, the patient's body should rest for 30 min. Afterward, Ipilimumab (Yervoy) (50 mg/10 mL) is administered at 3 mg/kg through intravenous infusion for 1 h and 30 min. The treatment should be repeated every 3 weeks for up to 6 doses or until unacceptable toxicity. Moreover, every 2 weeks, 240 mg of Nivolumab (Opdivo) should be administered via intravenous infusion for 6 months. |

## 4. Results

### 4.1. Clinical Cases Results

4.1.1. Lymphatic Nodules

The PET-CT and Ct scan examinations performed after the cryotherapy (Figure 7) showed stable activity of the hypermetabolic lymph node in the anterior mediastinum, a decrease in the activity of the hypermetabolic mass previously demonstrated in the anterior segment of the upper lobe of the left lung, which remains the same size.

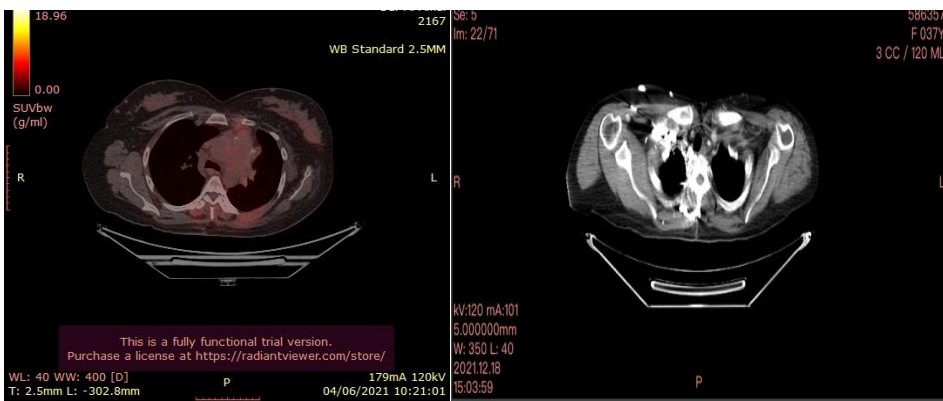

**Figure 7.** PETCT (**left**) and CT scan (**right**) images of the lymph node were performed after cryotherapy of the 35-year-old female patient.

There was also a disappearance of the hypermetabolic thickening of the left posterior pleura. Hypermetabolic lytic bone lesions were seen involving the right proximal femur T-10 and the left transverse process of T-9, regarding bone metastases in addition to subtle sclerotic changes noted related to healing.

In the follow-up checks, which continue to date occurring every 6 months for the first 3 years and then every 12 months, no local or distant metastatic repetitions have been highlighted and the patient is in apparent good health.

4.1.2. Lung Cancer

- Small Cell Lung Cancer (SCC)—Female Patient

The patient performed a follow-up with PET-CT after 6 months which showed no pleural or pericardial effusion, nor ilio-mediastinal lung tumor and axial or supraclavicular adenopathies. These were only small ilio-mediastinal adenopathies and a slight mediastinal thickening (Figure 8).

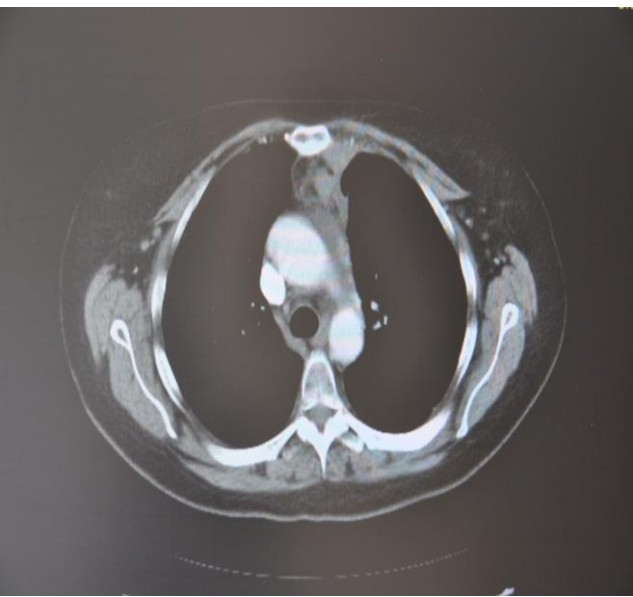

**Figure 8.** CT scan image, 6 months after treatment with cryotherapy and immunotherapy of the female patient with SCC.

In the follow-up checks, which continue to this day, that occur every 6 months for the first 3 years after cryosurgery and then every 12 months, the patient appears to be in an apparent state of good health.

- Small Cell Lung Cancer (SCC)—Male Patient

A follow-up CT scan after 6 months was done which showed no active lesion nor hilar mediastinal lung tumor or axillary or supraclavicular adenopathy. It consisted only of small ilio-mediastinal adenopathies and a slight mediastinal thickening (Figure 9). In the follow-up checks, which continue to this day occurring every 6 months for the first 3 years after cryosurgery and then every 12 months, the patient is in a good state of health.

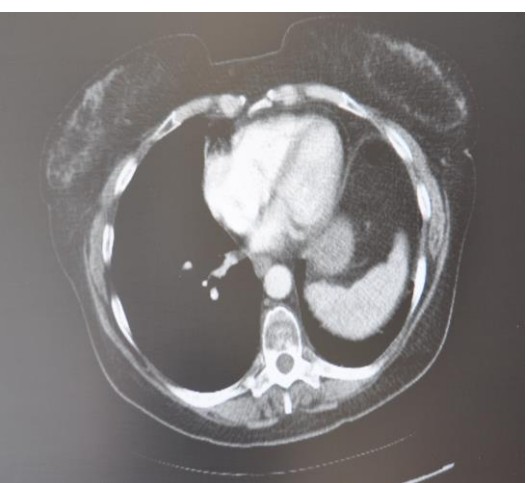

**Figure 9.** CT scan image, 6 months after treatment with cryotherapy and immunotherapy of the male patient with SCC.

### 4.1.3. Lung Cancer with Bone Metastases

After three months of the cryoablation, specifically in November 2021, a control CT scan and Pet-CT was performed. An impressive reduction in the size of the tumor is noted (Figure 10).

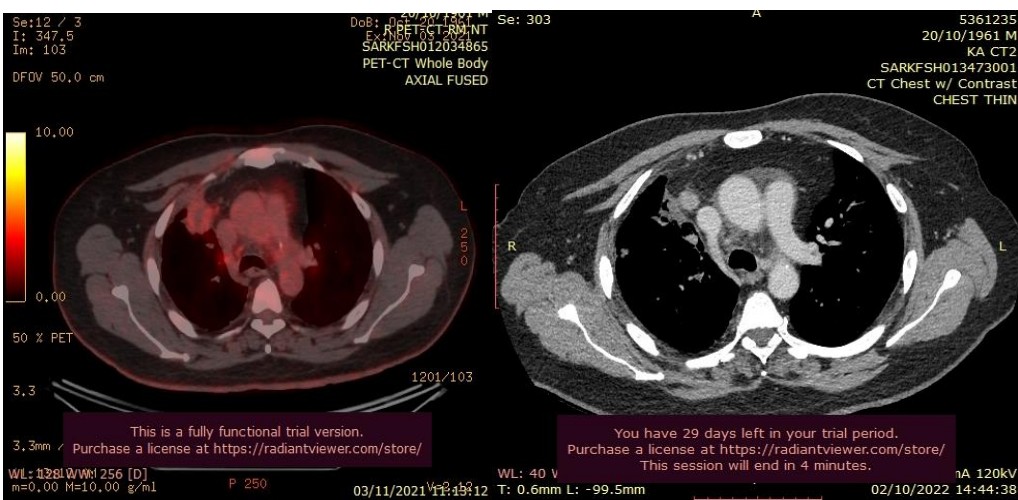

**Figure 10.** PET-CT (**left**) and Ct scan(**right**) images after cryotherapy in the patient with lung cancer accompanied by bone metastases.

In the follow-up checks, which continue to date and that occur every 6 months for the first three years after cryosurgery and then every 12 months, no local or distant metastatic repetitions have been highlighted and the patient appears to be in an apparent state of good health.

### 4.1.4. Bilateral Lung Metastasis and Mediastinal Lymph Nodes (Near the Esophagus)

After cryotherapy and immunotherapy, very good postoperative evolution was noted, both radiological and analytical, and clinical (Figure 11).

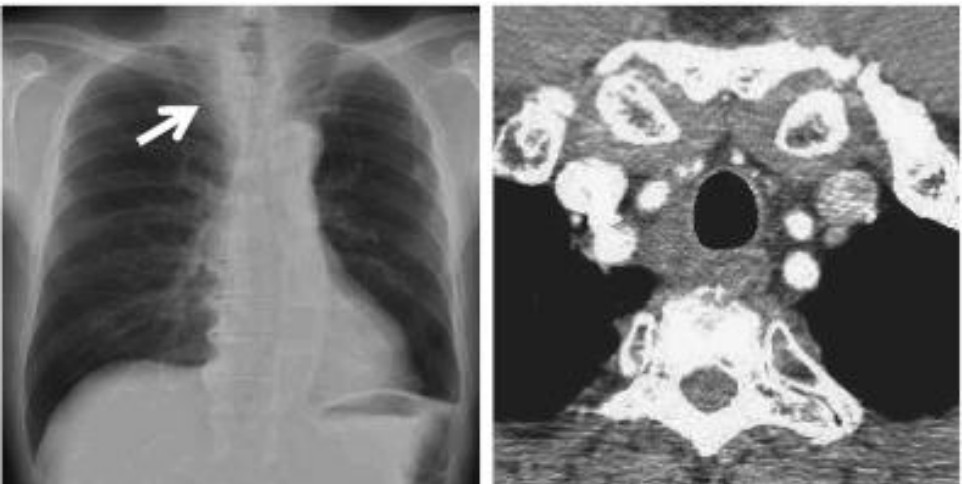

**Figure 11.** CT scan images after cryotherapy in the patient with bilateral lung metastasis and mediastinal lymph nodes. The arrow indicates where the tumor was located.

In the follow-up checks that occur every 6 months for the first 3 years after cryosurgery and then every 12 months, no local or distant metastatic repetitions have been highlighted and the patient appears to be in an apparent state of good health.

## 5. Discussion

The use of cryosurgery for tumor ablation is expanding, mainly due to its technical ease. Cryosurgery is a treatment that uses liquid nitrogen or argon gas to produce extreme colds to kill cancer cells and abnormal tissue. Because it is a local treatment, it is targeted at a specific area of the body [53]. Cryosurgery is used to treat cancers on the skin, as well as

some cancers inside the body such as lymph node, renal, and lung cancer, but also to treat metastases such as lung cancer metastasis. A cryo-immunologic response, which is the generation of an anti-tumor immune response triggered by the natural uptake of malignant tissue, is a potential secondary benefit of in situ freezing of malignant disease [54].

In the present research, five clinical cases of lymph node, lung cancer, bone, and lung metastasis were followed and analyzed.

Percutaneous cryoablation of the studied tumors was technically possible in this small series of patients. There does not appear to be any radiological evidence of new tumor development in follow-ups identical to that found by [55].

With respect to lung cancer cases, the most important series of lung cryoablation was published by [48] who accounted for the overall survival of 187 patients with 234 lung lesions following the use of cryoablation. Therefore, it can be confirmed that our case studies are in line with the study done by [48] where it is explicitly stated that lung cryoablation but also lymph node cryoablation modulated antitumor immunity to some extent, highlighting the immunological effect of cryotherapy. Cryoablation destroys tumor cells, releasing tumor antigens taken up by immature myeloid dendritic cells (DCs) take up. After cryoablation, DCs mature and migrate to the tumor-draining lymph nodes (TDLN), where they activate tumor-specific T cells that migrate to the tumor site, resulting in necrotic cell death and the production of inflammatory cytokines. Moreover, dying cancer cells might release "danger signals," such as nuclear proteins and pro-inflammatory cytokines, that attract and activate natural killer (NK) cells, macrophages, and neutrophils, according to [56,57].

Moreover, the health stability of the patient in our case studies after cryotherapy of lung and lymph node metastases was consistent with the result of [51] who concluded that the cryosurgery procedure of lung metastases was found to be a harmless method of attempting both tumor destruction and possibly specific immunological stimulation.

Regarding the influence of gender on the results of cryotherapy, no difference was noted. Both patients (female and male) had a positive response to cryotherapy. This concept is generally in all clinical cases but especially in the clinical case of SCC lung cancer, where two patients of different genders, having almost the same age (59 vs. 60), suffering from the same tumor and symptoms, treated with the same therapies (cryotherapy and immunotherapy), have had a positive response to these therapies.

Additionally, it's worth highlighting the positive role of cryoablation in renal cell carcinoma (RCC) and endophytic renal masses. Since the incidence of RCC is lately increasing in the elderly population, minimally invasive therapies such as cryoablation are being now preferred to enhance health-related quality of life [58]. Recent studies, such as [58–61], favored the percutaneous cryoablation over the partial nephrectomy in RCC stage cT1, especially for older people with low physical and role functional ratings and those who cannot undergo a surgical procedure. As well, [59,61–63] demonstrated the role of cryoablation in reducing complications and mortality, improving the oncological results and renal function after surgery, especially in the case of endophytic renal masses [62]. Furthermore, ref. [61] stressed the fact cryoablation showed 100% technical success and 5-year cancer-specific survival, especially for tumors larger than 3 cm.

Furthermore, in our clinical cases, after finalizing the cryoablation of the tumors, an immunotherapy treatment was administered mainly using the antibody drugs "Ipilumimab" and "Nivolumab" as inhibitors of the cytotoxic protein associated with T lymphocytes Associated protein-4 (CTLA-4) and Programmed cell Death protein-1 (PD-1) antagonists. Immune checkpoint inhibitor (ICI) mechanisms play a central role in immune tolerance, and tumor cells use these pathways to avoid immune attacks. CTLA-4 and PD-1 are checkpoint molecules that are predominantly expressed in T cells. By blocking CTLA-4 interaction sites, CD80/86 ligands become accessible for CD28 to activate T cells [64,65]. CTLA-4 antibodies also promote intratumorally regulatory T cell (Treg) depletion via the Fc receptor via antibody-dependent cell-mediated cytotoxicity (ADCC), owing to elevated CTLA-4 expression in tumor-infiltrating Tregs [66,67]. Tumors use the PD-1 signaling

pathway to produce T cell exhaustion, which is characterized by the co-expression of PD-1, TIM3, and LAG3 molecules [64]. Malignancies and tumor-associated macrophages (TAMs) express PD-L1 and play an important role in T cell exclusion from tumors [68]. Antibodies that inhibit PD-1 or PD-L1 are designed to reactivate fatigued T cells. Antibodies that block PD-1 or PD-L1 are intended to reawaken tired T cells. Combining CTLA-4 and PD-1 inhibition enhances survival in patients with melanoma, renal cell carcinoma, and a variety of other cancers. After receiving cryoablation and immunotherapy via ICI transfers, patients with lung, renal cell, and hepatocellular tumors saw synergistic benefits in terms of survival and immune responses [28,69,70]. This is consistent with our study since the results of the follow-ups to date have demonstrated the good health of the patients and the non-recurrence of the tumors. This also confirms the compatibly with the work of [71], the complementary roles of CTLA-4 and PD1 in the regulation of adaptive immunity, demonstrating that local immunotherapy with monoclonal antibodies enhanced the immunotherapeutic effects of cryosurgery alone and also provided an immune response to distant lesions. This phenomenon is known as the abscopal effect. Similarly, ref. [72] showed that immune checkpoint inhibitor (ICI) combination therapy produced an immune microenvironment in adjuvant and neoadjuvant settings of metastatic renal cancer.

## 6. Conclusions

It can be concluded that when cryoablation is combined with other immunomodulatory therapies, the immunological effect of cryoablation could be enhanced. Refs. [31,73,74] carried out studies related to cryoablation and immunotherapy showing favorable effects for cryoablation when combined with other therapies, enhancing the response anticancer immunity. This underscores the potential benefit of this ablation technique as an adjunct to immunotherapy. In accordance with what was discussed above, we believe that the excellent clinical and radiological response of our clinical cases of lymph nodes and lung cancer with and without bone metastases is an effect of cryosurgery and local immunotherapy.

## 7. Limitations of the Study

One of the main limitations of this study is that RECIST criteria were not applied to evaluate the efficacy of the treatment. Other limitations are related to the presence of a small number of patients and a brief follow-up period for evaluating local tumor control. However, these preliminary findings are indeed encouraging, and in light of our experience with this study, a larger, more thorough investigation is actively enrolling patients.

**Author Contributions:** Conceptualization, Z.a.A.M., F.L. and W.M.; methodology, Z.a.A.M., F.L., W.M. and S.S.; validation, S.S., W.M., P.T., A.C., A.U. and A.J.G.; formal analysis, F.L. and Z.a.A.M.; investigation, F.L. P.T., A.C., A.U. and A.J.G. resources, Z.a.A.M.; data curation, W.M. and F.L.; writing—original draft preparation, Z.a.A.M. and F.L.; writing—review and editing, F.L.; visualization, S.S., W.M., F.L., Z.a.A.M., P.T., A.C., A.U. and A.J.G.; supervision, F.L.; project administration, F.L. All authors have read and agreed to the published version of the manuscript.

**Funding:** This research did not receive any specific grant from funding agencies in the public, commercial, or not-for-profit sectors.

**Institutional Review Board Statement:** The study was conducted in accordance with the Declaration of Helsinki and approved by the Institutional Review Board (or Ethics Committee) of Hospital Hotel Dieu de France (protocol code 25/2018 and date of approval May 2018).

**Informed Consent Statement:** Informed consent was obtained from all subjects involved in the study.

**Data Availability Statement:** The data presented in this study are available on request from the corresponding author.

**Acknowledgments:** The authors thank Maria Sabbagh for her insightful comments from the first draft of this article as well when it was under review.

**Conflicts of Interest:** The authors declare no conflict of interest.

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
