# Peer review of "Cryoablation and Immunotherapy: An Enthralling Synergy for Cancer Treatment"

_curroncol, doi:10.3390/curroncol30050365_

Round 1

Reviewer 1 Report

In this manuscript the authors sought out ot clarify the the mutual relationship between therman (cryo) therapy ablation and ICIs. Overall interestin results. As an emergint topic, I would further suggest to add a pragraph in the Discussion section about the increasing role of thermal ablation in renal cell carcinoma (RCC) also considering this specific setting with the recent approval by FDA of ICIs (Pembro) in the neo and adjuvant setting. The authors should refer to (doi: 10.1016/j.euros.2022.09.013; doi: 10.3390/medicina58081041; doi: 10.1097/MOU.0000000000000868; doi: 10.3390/diagnostics13030388; doi: 10.21873/invivo.12413).

Limitations need to be stressed in the dedicated paragraph. Usually in the context of progression to metastatic disease and to evaluate the efficacy of a treatment the RECIST criteria are adopted. If RECIST criteria were not availabile please highlight this aspect as one of main limitation.

Author Response

Response to Reviewer 1 Comments

Point 1: As an emergint topic, I would further suggest to add a pragraph in the Discussion section about the increasing role of thermal ablation in renal cell carcinoma (RCC) also considering this specific setting with the recent approval by FDA of ICIs (Pembro) in the neo and adjuvant setting. The authors should refer to (doi: 10.1016/j.euros.2022.09.013; doi: 10.3390/medicina58081041; doi: 10.1097/MOU.0000000000000868; doi: 10.3390/diagnostics13030388; doi: 10.21873/invivo.12413).

Response 1: Thank you for your suggestions. We inserted a paragraph relatibe to the role of cryoablation in renal cell carcinoma (page 16 line 521 to 532). As well, we included as suggested the consideration of ICI in the neo and adjuvant setting (page 16 lines 536 to 552 and lines 559 to 561 ).

Point 2:  Limitations need to be stressed in the dedicated paragraph. Usually in the context of progression to metastatic disease and to evaluate the efficacy of a treatment the RECIST criteria are adopted. If RECIST criteria were not availabile please highlight this aspect as one of main limitation.

Response 2: Thank you for the pertinent comment. The RECIST criteria was not available this is why as suggested we included it in the limitations paragraph (page 17 line 577-579).

Reviewer 2 Report

Dear Authors,

Thank you for giving me the opportunity to review this interesting manuscript. Immunotherapy application in cancer treatment is being an emerging topic, in the recent period

This paper is original and well-designed, although some improvements should be done

- In methods:

The description of the clinical cases should be resumed in the main text. a table with a schematization of each clinical case could make the text more fluid

Could the pre and post-cryoablation treatment CT- scan imagines be inserted?

- In results:

The time of follow-up of clinical cases should be reported

Immunotherapy treatment is associated with peri-lesion inflammatory reactions. Did the authors observe differences in CT-scan imagines after cryoablation therapy in patients treated with immunotherapy, when compared to patients without immunotherapy treatment?

In discussion:

The biological effects of immunotherapy to improve cryoablation effect should be more extensively discussed

Author Response

Response to Reviewer 2 Comments

Point 1:

In methods:

  1. The description of the clinical cases should be resumed in the main text.

Response 1.A: Thank you for the comment. As suggested, we presented the theoretical description of the cancer types relative to the clinical cases in a new section ( 3.2. Clinical cases description) ( lines 170 to 207) and then in the 3.3. clinical cases presentation section we presented the cases that we treated.

  1. a table with a schematization of each clinical case could make the text more fluid.

Response 1.B: Thank you. We inserted the table showing the cancer type, the patient age and gender (if indicated), the admission date, the Ct, Pet and biopsy results prior to cryosyrgery and the immunotherapy treatment ( page 11, table 1).

  1. Could the pre and post-cryoablation treatment CT- scan imagines be inserted?

Response 1.C: We indicated for all the the pre and post cryoablation Ct – scan images as requested and when available also the pet CT images.

Point 2:  

In results:

  1. The time of follow-up of clinical cases should be reported

Response 2.A: Thank you for the suggestion. We inserted the follow-up time ( lines: 448-449, 462-463, 476-477).

  1. Immunotherapy treatment is associated with peri-lesion inflammatory reactions. Did the authors observe differences in CT-scan imagines after cryoablation therapy in patients treated with immunotherapy, when compared to patients without immunotherapy treatment?

Response 2.B: We can note inflammatory reactions in patients treated with immunotherapy and without immunotherapy because already cryoablation causes a cell injury leading to necrosis and vascular injury. So on, cryoablation with the combination of immune therapy can enhance innate immunity and tumor specific T lymphocyte activity, to help the immune system fight against cancer and to reduce as well the inflammatory lesions. This is why we believe that cryoablationh as the very good synergy with immune-therapy.

Point 3:  

In discussion:

The biological effects of immunotherapy to improve cryoablation effect should be more extensively discussed.

 Response 3: Thank you for the pertinent comment. We inserted a more extensive part regarding the effects of immunotherapy in order to improve cryoablation effect in the pages 16 (lines 536-552).

Reviewer 3 Report

The authors presented their case series investigating the ability of cryoablation to create a robust antitumor response in combination with immunologic agents. The current study hypothesized that this combination would result in a synergistic effect against cancer cells. The manuscript needs revision.

  • (The paper is organized as follows. In Section 2, we briefly report the methodological 71 approach and present the different cases with the relative therapy. Section 3 presents 72 results. In Section 5, we discuss the results and present the conclusions. Finally, in 73 Section 7 we examine some research limitations). This should be included in the methodological section.
  • The use of cryosurgery for tumor ablation is expanding, mainly due to its technical ease. 
  • All the references should be listed similarly (please check and fix all the references in the discussion).
  • The conclusion should be listed in a separate paragraph.
  • Figure 1 (Cryocare machine) should be eliminated because 
  • When presenting current indications for Cryoablation please consider including renal cell carcinoma management as it is listed in the current European and American Guidelines. For this purpose include this recent comparative study vs surgery (10.1089/end.2022.0478)
  • As different ablative treatments are currently available (MWA and PTA), I believe it is worthy of interest to describe them and discuss their differences and similitudes. For the scope consider including the following paper: 10.23736/S2724-6051.22.05092-3
  • Check typos.

Author Response

Response to Reviewer 3 Comments

Point 1: (The paper is organized as follows. In Section 2, we briefly report the methodological approach and present the different cases with the relative therapy. Section 3 presents results. In Section 5, we discuss the results and present the conclusions. Finally, in Section 7 we examine some research limitations). This should be included in the methodological section.

Response 1: Thank you for the suggestion. We included this paragraph at the beginning of the methodological part (lines 122-125).

Point 2:  All the references should be listed similarly (please check and fix all the references in the discussion).

Response 2: Thanks a lot. We fixed all the references in the discussion and throughout the text.

Point 3:    The conclusion should be listed in a separate paragraph.

Response 3: Thank you. We inserted the conclusion in a separate paragraph (page 17 line 566).

Point 4:  Figure 1 (Cryocare machine) should be eliminated.

Response 4: Thanks. We eliminated figure 1 as suggested and its relative sentence ( Page 4 line 141).

Point 5:  When presenting current indications for Cryoablation please consider including renal cell carcinoma management as it is listed in the current European and American Guidelines. For this purpose include this recent comparative study vs surgery (10.1089/end.2022.0478)

Response 5: Thank you for your significant comment. We included a paragraph regarding the renal cell carcinoma management as proposed by reviewer 1 and we added other pertinent information seen in the article proposed by you ( page 16 lines from 521 to 531).

Point 6:  As different ablative treatments are currently available (MWA and PTA), I believe it is worthy of interest to describe them and discuss their differences and similitudes. For the scope consider including the following paper: 10.23736/S2724-6051.22.05092-3.

Response 6: We thank you for this comment. We included a new section (section 2: Overview on the current ablation techniques) which is relative to the three current ablative treatments (MWA, PTA and RFA). We described each technique inserting its advantages and disadvantages, as well as the differences and similitudes compared to the other techniques (page 2 lines from 75 to 90, page 3 lines from 95 to 122).

Point 7: Check typos.

Response 7: Thank you for noting that. We check the different typos (lines 19, 23, 58, 76, 80, 93, 105, 117, 339, 378).

Round 2

Reviewer 1 Report

The authors revised the manuscript fully and properly. No further comments.

Reviewer 3 Report

the revised version presents significant improvements and it is worthy for publication.